# Antiviral Activities of Ethyl Pheophorbides a and b Isolated from *Aster pseudoglehnii* against Influenza Viruses

**DOI:** 10.3390/molecules28010041

**Published:** 2022-12-21

**Authors:** Subin Park, Ji-Young Kim, Hak Cheol Kwon, Dae Sik Jang, Yoon-Jae Song

**Affiliations:** 1Department of Life Science, Gachon University, Seongnam 13120, Republic of Korea; 2Department of Biomedical and Pharmaceutical Sciences, Graduate School, Kyung Hee University, Seoul 02447, Republic of Korea; 3Natural Product Informatics Research Center, Korea Institute of Science and Technology (KIST), Gangneung Institute, Gangneung 25451, Republic of Korea

**Keywords:** *Aster pseudoglehnii*, endemic plants, ethyl pheophorbide, chlorophyll derivative, virucidal

## Abstract

Screening of the antiviral and virucidal activities of ethanol extracts from plants endemic to the Republic of Korea revealed the inhibitory activity of a 70% ethanol extract of the whole plant of *A. pseudoglehnii* (APE) against influenza virus infection. Two chlorophyll derivatives, ethyl pheophorbides a and b, isolated as active components of APE, exerted virucidal effects with no evident cytotoxicity. These compounds were effective only under conditions of direct incubation with the virus, and exerted no effects on the influenza A virus (IAV) surface glycoproteins hemagglutinin (HA) and neuraminidase (NA). Interestingly, virucidal activities of ethyl pheophorbides a and b were observed against enveloped but not non-enveloped viruses, suggesting that these compounds act by affecting the integrity of the viral membrane and reducing infectivity.

## 1. Introduction

Influenza viruses belong to the Orthomyxoviridae family and are divided into A, B, and C types [1]. Among these, influenza B virus (IBV) and influenza C virus (ICV) specifically infect humans, whereas influenza A virus (IAV) is a major pathogen responsible for global pandemics owing to its zoonotic transmission. Influenza viruses display pathogenicity as well as genetic and structural differences. However, similar structures for IAV and IBV have been reported [2,3]. Their genomes contain eight negative-sense, single-stranded RNA fragments encoding more than ten proteins [4]. The eight RNA fragments are surrounded by nucleocapsid protein (NP) and form a ribonucleoprotein (RNP) complex together with RNA polymerase complex proteins (PA, PB1, PB2). RNP is surrounded by matrix protein (M1) and the nucleocapsid is encapsulated by a cell membrane-derived phospholipid bilayer. Hemagglutinin (HA) and neuraminidase (NA), spike glycoproteins that determine pathogenicity, are located on the surface of the virus membrane, while ion channels span the membrane [5,6]. NA and HA proteins are divided into subtypes according to their structural and genetic characteristics. So far, 16 subtypes for HA and 9 subtypes for NA have been classified, among which HA1, 2, and 3 and NA1 and 2 are contributory factors to epidemic and pandemic outbreaks [4,7,8]. Recent studies have revealed H1N1 and H3N2 of IAV and IBV as causative agents of seasonal flu [9]. Other important viral proteins include nonstructural protein 1 (NS1) and nonstructural protein 2 (NS2). NS1 functions to evade the immune response of cells, and NS2 is a nuclear export protein (NEP) that plays a key role in nuclear release of RNP [10].

Influenza virus infects approximately 5 million patients, resulting in 250,000–500,000 deaths each year [8]. Over the past hundred years, four pandemics have occurred, specifically, H1N1 Spanish flu in 1918, H2N2 Asian flu in 1957, H3N2 Hong Kong flu in 1968, and H1N1 swine flu in 2009. The 1918 H1N1 influenza pandemic caused an estimated 60 million deaths and is recorded as one of the most deadly events in human history [4,11]. The virus is mainly transmitted through the respiratory system and can spread through air, feces and other secretions. Common symptoms include fever, muscle pain, headache, cough, and sore throat, which generally last for 7–10 days. While no special treatment is required, patients are likely to develop severe symptoms or complications [12,13,14,15].

Numerous antiviral drugs have been developed against influenza virus that target the ion channel (M2), NA, RNA polymerase, and inosine 5′ monophosphate (IMP) dehydrogenase proteins. Among these, only antiviral agents targeting M2 and NA have been approved by the FDA [12,15]. Amantadine and rimantadine were originally developed as M2 inhibitors, but their use is limited since these compounds only affect IAV and are associated with resistance or neurological side-effects [13,14]. Sialic acid analogs targeting NA, oseltamivir and zanamivir, were subsequently developed [16]. However, these antiviral agents were also prone to side-effects and resistance issues over time [17,18]. The influenza virus modifies its HA and NA proteins through antigenic drift and shift to avoid cellular immune responses [19,20,21]. Therefore, novel antiviral agents with a new mechanism against these mutations, broad-spectrum activity against both IAV and IBV, and no adverse effects are an urgent clinical requirement.

Natural products contain chemical constituents with various biological functions including antiviral activities. In particular, edible plants are emerging as valuable sources of antiviral agents because of their low toxicity [22,23]. Various natural compounds isolated from edible plants including chlorogenic acids, diarylheptanoids, flavonoids, flavonoid glycosides, iridoids and lignans exhibited antiviral activities against influenza viruses [24]. *A. pseudoglehnii* (Compositae), a perennial plant native to Ulleung Island, is an endemic species to the Republic of Korea [22]. Young leaves and stems of *A. pseudoglehnii* are a common ingredient in Korean cuisine. This plant is traditionally used as a cold antipyretic, expectorant for tonsillitis, and antitussive agent. In addition, anti-adipogenic, anti-obesity, anti-oxidant, and anti-inflammatory effects of ethanol extracts of *A. pseudoglehnii* have been reported [25,26,27]. In this study, we focused on the virucidal activities of bioactive compounds isolated from a 70% ethanol extract of whole plants of *A. pseudoglehnii* (APE) against IAV.

## 2. Results

### 2.1. Virucidal Activity of APE against IAV

To examine the anti-influenza virus activity of APE, viruses and cells were subjected to a time-of-drug-addition assay as shown in Figure 1A. In the first method, IAV was treated with the extract for 1 h and Madin-Darby canine kidney (MDCK) cells were infected with the mixture. At 1 h after infection, the mixture was removed (pre-treatment). In the second method, IAV was treated with the extract and MDCK cells were immediately infected with the mixture. At 1 h after infection, the mixture was removed (adsorption). In the third group, MDCK cells were infected with IAV which was removed after 1 h. Cells were subsequently treated with the extract (post-infection). At 24 h after infection, viral RNA transcript levels were evaluated via qRT-PCR. DMSO treatment was conducted as a control. Treatment with the concentrations up to 100 µg/mL of APE exhibited no significant cytotoxic activity against MDCK cells. Compared to the control, IAV replication was significantly inhibited in the pre-treatment group (Figure 1B), confirming its virucidal activity against IAV.

### 2.2. Isolation and Identification of Ethyl Pheophorbides a and b

To identify the bioactive components with virucidal activity against IAV, APE was subjected to repeated column chromatography. Two chlorophyll derivatives were isolated from the active *n*-hexane subfractions HE5 and HE6 (Figure 2). Compound **1** was isolated as a green oil and its molecular formula established as C_37_H_40_N_4_O_5_ via high-resolution quadruple time-of-flight mass spectrometry (HR-Q-TOF-MS) (*m*/*z* = 621.3266 [M + H]^+^; calculated for C_37_H_41_N_4_O_5_, 621.3077) (Appendix A). The ^1^H NMR spectrum of **1** displayed signals for a group of vinylic hydrogens at *δ*_H_ 6.16 (1H, dd, *J* = 11.5, 1.5 Hz, H-3^2^a), 6.27 (1H, dd, *J* = 18.0, 1.5 Hz, H-3^2^b), and 7.96 (1H, dd, *J* = 18.0, 11.5 Hz, H-3^1^), three olefinic methyls at *δ*_H_ 3.20 (3H, s, H-7^1^), 3.38 (3H, s, H-2^1^), and 3.67 (3H, s, H-12^1^), and three olefinic methines at *δ*_H_ 8.56 (1H, s, H-20), 9.36 (1H, s, H-5), and 9.50 (1H, s, H-10) (Appendix A and Table 1). Additionally, methoxylic hydrogen at *δ*_H_ 3.86 (1H, s, H-13^4^) and ethoxylic hydrogens at *δ*_H_ 1.09 (3H, t, *J* = 7.0 Hz, H-17^5^) and 4.00 (2H, m, H-17^4^) were observed. The ^13^C NMR spectrum of **1** contained 37 signals including six methyl carbons (*δ*_C_ 11.2, 12.1, 12.2, 14.1, 17.4, and 23.1), one methoxy group (*δ*_C_ 52.9), four olefinic groups (*δ*_C_ 93.2, 97.5. 104.5, and 129.2), and an exomethylene group (*δ*_C_ 122.9), inferring that **1** is similar to the porphyrin nucleus of pheophytins except for the presence of an ethyl group (Appendix A and Table 1) [28]. Compound **1** was identified as ethyl pheophorbide a based on spectroscopic evidence and comparison of data from the literature [29].

Compound **2** was also isolated as a green oil. Data from LC-ESI-MS revealed an [M + 1]^+^ peak at *m*/*z* 621.4 (Appendix A). The ^1^H NMR spectrum of **2** displayed four singlets at *δ*_H_ 8.55 (1H, s, H-20), 9.68 (1H, s, H-10), 10.31 (1H, s, H-5), and 11.17 (1H, s, H-7^1^) in the low-field region (Appendix A and Table 1). Moreover, a terminal vinyl group [*δ*_H_ 6.24 (1H, dd, *J* = 11.5, 1.0 Hz, H-3^2^a), 6.38 (1H, dd, *J* = 18.0, 1.0 Hz, H-3^2^b), and 8.02 (1H, dd, *J* = 17.5, 11.5 Hz, H-3^1^)], two ethyl groups [*δ*_H_ 1.12 (3H, t, *J* = 6.5 Hz, H-13^5^), 1.84 (3H, t, *J* = 7.5 Hz, H-8^2^), 3.90 (2H, q, *J* = 7.5 Hz, H-8^1^), and 4.46 (2H, q, *J* = 7.0 Hz, H-13^4^)], and three methyl signals [*δ*_H_ 1.84 (3H, d, *J* = 7.5 Hz, H-18^1^), 3.39 (3H, s, H-2^1^) and 3.70 (3H, s, H-12^1^)] were observed. ^1^H and ^13^C NMR spectra of **2** were very similar to those of **1** (Appendix A and Appendix A). The ^13^C NMR spectrum exhibited 36 carbon signals and the presence of an aldehyde was additionally confirmed at 187.8 ppm. Based on spectroscopic evidence and comparison with earlier literature, compound **2** was identified as ethyl pheophorbide b [30].

### 2.3. Virucidal Activities of Ethyl Pheophorbides a and b against Influenza Virus

To determine the antiviral activities of ethyl pheophorbides a and b against IAV, viruses were treated with the compounds and experiments were performed as shown in Figure 1A. Amantadine, which inhibits the M2 ion channel and, in turn, virus uncoating, was used as a control. Similar to the results obtained with APE, IAV transcript levels were significantly reduced upon pre-treatment of the virus with both compounds, while amantadine treatment during virus adsorption to host cells induced a marked decrease in IAV replication (Figure 3A). The IAV HA protein level was further determined with western blot analysis (Figure 3B). Notably, expression of HA protein was significantly suppressed upon pre-treatment of viruses with ethyl pheophorbides a and b (Figure 3B; lines 3 and 8). Amantadine treatment during adsorption also induced marked downregulation of HA protein (Figure 3B; lines 13 and 14). Amantadine plays an important role in the fusion of viral and endosomal membranes during influenza virus entry into host cells, while ethyl pheophorbides a and b may affect virus infectivity before entry into host cells.

To further ascertain whether ethyl pheophorbides a and b exhibit inhibitory effects against IBV, IAV and IBV were pre-treated with different concentrations of each compound for 1 h at room temperature, followed by infection of MDCK cells with the mixture containing IAV and IBV at MOI of 0.01 and 0.001, respectively. At 1 h after infection, the mixture was removed and viral titers were determined using the plaque reduction assay at 48 h after infection. Pre-treatment with ethyl pheophorbides a and b led to a significant reduction in the IAV and IBV plaque numbers (Figure 4), with estimated 50% inhibitory concentrations (IC_50_) of 0.086 ± 0.072 µg/mL and 0.094 ± 0.058 µg/mL for IAV (Figure 4A), and 1.62 ± 0.86 µg/mL and 3.17 ± 0.94 µg/mL for IBV (Figure 4B), respectively.

To clarify whether the inhibitory activities of ethyl pheophorbides a and b are mediated through cytotoxicity, MDCK cells were treated with different concentrations of each compound and cell viability was determined after 48 h via the MTT assay (Appendix A). Both ethyl pheophorbides exerted no adverse effects on cell viability up to a concentration of 10 µg/mL, indicative of virucidal activity against influenza viruses without cytotoxic effects.

### 2.4. Effects of Ethyl Pheophorbides a and b on Major Surface Glycoproteins of Influenza Virus

We further considered the possibility that ethyl pheophorbides a and b prevent viral binding to host cells by targeting the IAV major surface glycoproteins NA and HA. To explore whether ethyl pheophorbides a and b exert inhibitory effects on NA, IAV was treated with the compounds (1 µg/mL) and the NA inhibition assay was performed (Figure 5A). Zanamivir, a NA inhibitor was used as the control and significantly suppressed NA activity, as expected. Notably, ethyl pheophorbides a and b exerted no effects on IAV NA.

Next, to ascertain whether the two ethyl pheophorbide compounds inhibit HA, two assays using chicken red blood cells (cRBC) were employed. HA is a dimer composed of HA1 and HA2. HA1 contains a domain that binds sialic acid of cells and HA2 interacts with the cell membrane via structural rearrangement induced by low pH during fusion of the viral and cell membranes [31]. First, to establish whether the compounds target HA1, IAV was treated with ethyl pheophorbides a and b (1 µg/mL) and the hemagglutination inhibition assay was performed (Figure 5B). In blank samples devoid of IAV, no red blood cell aggregation was observed, whereas in samples of IAV treated with DMSO, ethyl pheophorbide a or ethyl pheophorbide b, the agglutination reaction occurred (Figure 5B). The potential inhibitory effects of ethyl pheophorbides a and b on HA2 were examined using the hemolysis inhibition assay (Figure 5C). To this end, cRBC (2%) was mixed with IAV treated with ethyl pheophorbides a and b (1 µg/mL), followed by treatment with sodium acetate-acetic acid (0.5 M, pH 5.2) to induce hemolysis via structural changes of HA2. Compared to the blank sample without IAV, relatively high absorbance was observed in samples of IAV treated with ethyl pheophorbide a or ethyl pheophorbide b (Figure 5C). Overall, ethyl pheophorbides a and b had no effect on the functions of HA1 and HA2, clearly suggesting inhibitory activity on infectivity of influenza viruses via a mechanism that does not involve major viral surface glycoproteins.

### 2.5. Effects of Ethyl Pheophorbides a and b on Enveloped Viruses

To determine whether ethyl pheophorbides a and b reduce infectivity of influenza virus via effects on the viral membrane, other enveloped viruses including hepatitis C virus (HCV), human cytomegalovirus (HCMV), Japanese encephalitis virus (JEV), and non-enveloped adenovirus were treated with the compounds. At 1 h after treatment, viral infectivity was determined using plaque assays for HCMV, JEV and adenovirus, or immunofluorescence assay for HCV. Both ethyl pheophorbide compounds significantly reduced replication of HCV, JEV and HCMV (Figure 6A–C). The IC_50_ values of ethyl pheophorbide a against HCV, HCMV and JEV were 3.04 ± 1.69 µg/mL, 0.45 ± 0.26 µg/mL, and 8.66 ± 1.44 ng/mL, respectively. The IC_50_ values of ethyl pheophorbide b against HCV, HCMV and JEV were 2.19 ± 1.12 µg/mL, 0.46 ± 0.30 µg/mL, and 10.17 ± 1.79 ng/mL, respectively. Ethyl pheophorbides a and b clearly inhibited all enveloped viruses but had no effect against non-enveloped adenovirus (Figure 6D). Based on the collective results, we propose that ethyl pheophorbides a and b exhibit broad-range virucidal activities against enveloped viruses, possibly by affecting the integrity of the viral membrane.

## 3. Discussion

*A. pseudoglehnii* is an edible plant endemic to the Republic of Korea with low reported toxicity and multiple beneficial pharmacological properties. In this study, pre-treatment of IAV with APE induced a virucidal effect and the chlorophyll derivatives ethyl pheophorbides a and b were isolated as bioactive chemical constituents.

Chlorophyll is responsible for absorbing light during photosynthesis in plants. However, excessive photosynthesis could function as a photosensitizer and cause damage to cells. Therefore, synthesis and degradation of chlorophyll is an important process in preventing cytotoxicity, and various intermediates are generated during chlorophyll degradation [32]. Ethyl pheophorbides a and b are among the intermediates that act as a photosensitizers on human cancer cells, showing anti-tumor efficacy [33]. In this study, ethyl pheophorbides a and b exhibited broad virucidal activities against enveloped viruses upon direct treatment before infection of host cells, supporting the theory that ethyl pheophorbides a and b affect the viral membrane as photosensitizers.

Chemical compounds isolated from plant extracts with virucidal activity against enveloped viruses have been previously reported. For instance, curcumin inhibits the activity of enveloped viruses by damaging the viral membrane structure, leading to increased membrane permeability. Upon insertion of curcumin into the phospholipid bilayer of the membrane, its phenol ring interacts with the hydrogen bond site of the membrane and induces structural changes [34]. Ethyl pheophorbides a and b are proposed to have a similar mechanism of action as curcumin. In addition, chlorophyll derivatives similar in structure to ethyl pheophorbides were reported to reduce the infectivity of enveloped viruses by acting as photosensitizers to mediate physicochemical changes [35]. Notably, while the membrane of influenza virus is derived from the cell membrane, the compounds were not toxic to cells and showed specific activity against enveloped viruses, which should be explored in further studies.

In this study, IC_50_ values of ethyl pheophorbide a against IAV and IBV were 0.085 and 1.02 µg/mL, respectively, and antiviral activity against IBV was relatively lower. The reasons underlying the distinct virucidal activities of ethyl pheophorbides a and b against different enveloped viruses require further investigation. Interestingly, however, examination of the anti-influenza virus activity of commercially purchased pheophorbide a devoid of an ethyl group revealed similar IC_50_ values against IAV and IBV (data not shown). We could not effectively compare the virucidal activities of commercially synthesized compounds with those of isolates from plant extracts. To establish whether the ethyl group serves as a critical moiety, it is necessary to synthesize these compounds to the same levels of purity and compare their virucidal effects.

## 4. Materials and Methods

### 4.1. Cells, Viruses and Plant Materials

Madin-Darby canine kidney (MDCK) cells, influenza A virus (PR8-GFP), and influenza B virus (wild-type B/Brisbane/60/2008) were provided by Professor Manseong Park (Korea university, Republic of Korea). MDCK cells were cultured in Modified Eagle’s medium (MEM) (Hyclone, Logan, UT, USA) supplemented with 10% fetal bovine serum (FBS) and 1X penicillin-streptomycin (1X P/S). IAV and IBV were propagated in fertilized chicken eggs. Specific pathogen-free chicken eggs (Orientbio, Kyunggi-do, Republic of Korea) were cultured for 10 days at 37 °C and 55–60% humidity. Viruses were diluted to 10^3^–10^4^ PFU/mL with buffer (1X PBS, 10 mM HEPES, pH 7.4) and injected into the allantoic cavity of eggs. After culturing in an incubator (48 h for IAV and 72 h for IBV), allantoic fluid was obtained and the virus titer was determined using the plaque assay. MDCK cells were inoculated with influenza virus, with addition of infection medium (MEM, 0.3% BSA, 1X P/S, 1 µg/mL TPCK-trypsin) after incubation at 37 °C for 1 h. Baby hamster kidney cells (BHK21) were purchased from the Korean Cell Line Bank (KCLB, Seoul, Republic of Korea) and Japanese encephalitis virus (JEV, genotype1) obtained from the National Culture Collection for Pathogens (NCCP, Chungcheong-do, Republic of Korea). BHK21 and 293A cells were cultured in Dulbecco’s modified Eagle’s medium (DMEM) (Hyclone) supplemented with 10% FBS and 1X P/S. BHK21 cells were inoculated with JEV at 37 °C for 1 h, followed by the addition of DMEM containing 2% FBS and 1X P/S. After 48 h, JEV was obtained from the supernatant. Recombinant adenovirus expressing GFP was generated by inserting the GFP gene into an adenoviral vector. The GFP gene was amplified using pEF-GFP vector (Addgene #11154) as a template. The primer sequences used were 5′-GGGGACAAGTTTGTACAAAAAAGCAGGCTTTATGGTGAGCAAGGGCGAGGAGCTGT-3′ (forward) and 5′-GGGGACCACTTTGTACAAGAAAGCTGGGTTTTACTTGTACAGCTCGTCCATGCCGA-3′ (reverse), and PCR was performed using 5X HOT FIREPol^®^ Blend Master Mix (Solis Biodyne, Tartu, Estonia). The amplified GFP gene was cloned using the Gateway system (with Gateway™ BP Clonase™ II and Gateway™ LR Clonase™ II enzyme mixtures and pAd/CMV/V5-DEST™ Gateway™ Vector for insertion of the GFP gene (Invitrogen, MA, USA)), followed by transfection into 293A cells, as described previously [36]. Upon ≥50% cell death, which was determined by observing cytopathic effects (CPE), freeze-thawing at −80 °C and 37 °C was conducted three times to obtain adenovirus expressing GFP from the supernatant. Protocols for the maintenance and propagation of human cytomegalovirus (HCMV) Towne strain, hepatitis C virus (HCV, JFH-1), human hepatocyte (Huh7.5) and primary human foreskin fibroblasts cells (HFF) have been described previously [37,38]. *Aster pseudoglehnii* Y. Lim, J. O. Hyun, and H. Shin (Compositae) was provided by Hantaek Botanical Garden (Kyunggi-do, Republic of Korea) in May 2020 (voucher specimen #: HTS2021-0001).

### 4.2. General Experimental Procedures

Silica gel (Merck 60A, 70–230 and 230–400 mesh ASTM; Merck, Kenilworth, MA, USA) was used for column chromatography. Pre-packed cartridges with Redi Sep-C18 columns (43 g; Teledyne Isco, Lincoln, NE, USA) were employed for flash chromatography. High-performance liquid chromatography (HPLC) was performed using a Gemini NX-C18 110A column (250 × 21.2 mm i.d. 5 μm; Phenomenex, Torrance, CA, USA). Flash chromatography was performed via flash purification (Combi Flash Rf, Teledyne Isco) on a Waters purification system (1525 pump, PDA 1996 detector; Waters, Milford, MA, USA). Prior to chromatographic separation, all solvents were distilled. Thin-layer chromatography (TLC) analyses were performed on Silica gel 60 F_254_ (Merck) and RP-18 F_254S_ (Merck) plates. Compounds were visualized under UV light (254 and 365 nm) by dipping plates into a 20% (*v*/*v*) H_2_SO_4_ reagent (Duksan, Kyunggi-do, Republic of Korea) and were then heated at 120 °C for 10-15 min. Nuclear magnetic resonance (NMR) spectroscopy was performed using a JNM-ECZ500R (JEOL, Tokyo, Japan) instrument with deuterium solvent (Cambridge Isotope Laboratories, Tewksbury, MA, USA) as an internal standard, and with chemical shifts expressed as *δ* values (ppm). Liquid chromatography-electrospray ionization mass spectrometry (LC–ESI–MS) was conducted using a Waters ACQUITY UPLC system and Waters Micromass Quattro micro API with an ACQUITY UPLC BEH C18 column (2.1 × 50 mm i.d. 1.7 μm, Waters). High resolution mass spectra (HR-MS) were obtained via quadrupole time-of-flight mass spectrometry (Q-TOF-MS) (Thermo Fisher Scientific Inc., Waltham, MA, USA).

### 4.3. Isolation of Ethyl Pheophorbides a and b from APE

Dried whole *A. pseudoglehnii* (966.5 g) plants were extracted twice with 70% EtOH for seven days at room temperature. The solvent was removed *in vacuo* to generate a 70% EtOH extract (136.07 g), which was suspended in 1 L of water and partitioned with *n*-hexane and *n*-butanol to obtain *n*-hexane- (8.38 g), *n*-butanol- (43.28 g), and water-soluble fractions (70.35 g), respectively. The *n*-hexane soluble fraction displayed the most potent activity in the anti-virus assay. Consequently, this fraction (7 g) was subjected to column chromatography (CC) (4.8 × 44.5 cm) using a silica gel (70–230 mesh) column and eluted with *n*-hexane-EtOAc (8:2 to 0:10, *v*/*v*) into ten subfractions (HE1–HE10). Among these subfractions, HE5 and HE6 showed significant anti-influenza virus activity.

HE5 (624.9 mg) was further fractionated using silica gel CC (230–400 mesh; 2.8 × 28.5 cm) with *n*-hexane–EtOAc (10:0 to 5:5, *v*/*v*) into five subfractions (HE5-1–HE5-5). HE5-3 (185.0 mg) was further separated using a flash chromatography system with Redi Sep-C18 (43 g, acetonitrile–water, from 7:3 to 10:0, *v*/*v*) into three subfractions (HE5-3-1–HE5-3-3). Compound **1** (ethyl pheophorbide a, 2.7 mg) was obtained via preparative HPLC with a Gemini 5μm NX-C18 110 A column (acetonitrile water = 9:1 to 10:0, *v*/*v*).

HE6 (112.8 mg) was subjected to chromatography on a silica gel (230–400 mesh; 2.8 × 25.7 cm) with n-hexane–EtOAc (10:0 to 5:5, *v*/*v*) into four subfractions (HE6-1–HE6-4). Preparative HPLC of HE6-2 (10.8 mg) with a Gemini 5μm NX-C18 110 A column (acetonitrile-water = 9:1 to 10:0, *v*/*v*) yielded compound **2** (ethyl pheophorbide b, 2.0 mg).

#### 4.3.1. Ethyl Pheophorbide a

Green oil; HR-Q-TOF-MS (positive mode) *m*/*z* = 621.3226 [M + H]^+^, ^1^H and ^13^C NMR data: Table 1.

#### 4.3.2. Ethyl Pheophorbide b

Green oil; LC-ESI-MS (positive mode) *m*/*z* = 621.4 [M + H]^+^, ^1^H and ^13^C NMR data: Table 1.

### 4.4. Time-of-Drug-Addition Assay

Viruses and cells were subjected to three different treatments. In the first method, IAV was treated with the extract for 1 h at room temperature and Madin-Darby canine kidney (MDCK) cells were infected with the mixture at a multiplicity of infection (MOI) of 0.01. At 1 h after infection, the mixture was removed (pre-treatment). In the second method, IAV was treated with the extract at room temperature and MDCK cells were immediately infected with the mixture at a MOI of 0.01. At 1 h after infection, the mixture was removed (adsorption). In the third group, MDCK cells were infected with IAV at a MOI of 0.01, which was removed after 1 h. Cells were subsequently treated with the extract (post-infection). At 24 h after infection, cells were harvested to extract total RNA, and viral RNA transcript levels were evaluated via qRT-PCR.

### 4.5. Plaque Reduction Assay

IAV and IBV were treated with different concentrations of ethyl pheophorbide a or b and incubated at room temperature for 1 h. MDCK cells were infected with the mixtures and the supernatant was obtained at 48 h after infection. MDCK cells were infected with serially diluted supernatant, and after 1 h, incubated with a 1:1 mixture of 2X DMEM (2X DMEM, 0.4% BSA, 25 mM HEPES, 1X P/S) and 2% agarose containing 1 µg/mL TPCK-trypsin. Adenovirus, HCMV and JEV were treated with different concentrations of ethyl pheophorbide a or b (0–10 µg/mL) and the mixtures were incubated at room temperature for 1 h. BHK21 cells were infected with serially diluted JEV for 1 h, followed by incubation in medium (DMEM, 2% FBS, 1X P/S) containing 0.5% agarose. At 48 h after incubation, cells were fixed with 10% formaldehyde at room temperature and stained with 0.3% crystal violet. 293A cells were infected with serially diluted adenovirs for 6 h, followed by incubation in medium (DMEM, 2% FBS, 1X P/S) containing 0.4% agarose. At 48 h after infection, medium containing 0.4% agarose was added. At 4 to 6 days after infection, Thiazolyl Blue Tetrazolium Bromide (MTT; Sigma-Aldrich, St. Louis, MO, USA) solution was added to the medium at a concentration of 0.5 mg/mL and plaques were observed 1 h after incubation at 37 °C. The plaque assay for HCMV has been described in a previous report [39]. The number of plaques was counted and half-maximal inhibitory concentration (IC_50_) values were calculated using GraphPad Prism 7 (GraphPad Software, San Diego, CA, USA).

### 4.6. Immunofluorescence Assay

Cells were fixed with a 1:1 mixture of methanol and acetone for 10 min at −20 °C, washed with 1X PBS, and blocked with 1% BSA at room temperature for 1 h. Subsequently, cells were incubated with primary antibody overnight at 4 °C, followed by secondary antibody at room temperature for 1 h. After staining of cells with 4,6-diamidino-2-phenylindole (DAPI) (Vector Laboratories, Burlingame, CA, USA), fluorescence was examined under a Nikon TS100-F fluorescence microscope (Tokyo, Japan) equipped with a digital camera. Fluorescence images were analyzed using Nikon NIS-Elements microscope imaging software. An antibody against HCV core protein was purchased from Anogen (Mississauga, ON, Canada). Alexa 488-labeled secondary antibody was purchased from Thermo Fisher Scientific (Waltham, MA, USA).

### 4.7. MTT Assay

MDCK cells were treated with different concentrations of ethyl pheophorbide a or b (0–10 µg/mL) for 48 h and subsequently incubated with 0.5 mg/mL MTT solution at 37 °C. After a 4 h period, tetrazolium crystals were dissolved with DMSO and absorbance was measured at 560 nm.

### 4.8. Western Blot Analysis

Cells were fractionated and transferred to nitrocellulose membrane as described previously [40]. Antibodies specific for IAV H1N1 HA and tubulin were purchased from Genetex (Irvine, CA, USA) and Sigma-Aldrich, respectively. A peroxidase-labeled anti-mouse/rabbit immunoglobulin G antibody was purchased from Jackson ImmunoResearch Laboratories (West Grove, PA, USA).

### 4.9. Quantitative Reverse Transcription PCR (qRT-PCR)

To determine viral transcripts levels, total RNA was extracted using a HiGene™ Total RNA Prep Kit (Biofact, Daejeon, Republic of Korea). RNA was reverse-transcribed using a TOPscript™ cDNA synthesis kit (Enzynomics, Chungcheong-do, Republic of Korea) to synthesize complementary DNA (cDNA), which was amplified using the StepOnePlus Real-Time PCR system (Applied Biosystems, Foster City, CA, USA) with HOT FIREPol^®^ EvaGreen qPCR mix Plus (Solis BioDyne) and specific primers. The primer sequences used for amplification were as follows: IAV M1, 5′-GACCAATCCTGTCACCTCTGAC-3′ (forward) and 5′-AGGGCATTTTGGACAAACCGTCTA-3′ (reverse); Canis lupus familiaris glyceraldehyde-3-phosphate dehydrogenase (GAPDH), 5′-CCAGGGCTGCTTTTAACTCTGG-3′ (forward) and 5′-ACTGTGCCGTGGAATTTGCCG-3′ (reverse).

### 4.10. NA Inhibition Assay

IAV was diluted to 8 × 10^5^ PFU using 1X Assay buffer (33.3 mM 2-(N-morpholino) ethanesulfonic acid (MES), 4 mM CaCl_2_, pH 6.5) containing 0.1% NP-40 and treated with ethyl pheophorbide a (1 µg/mL), ethyl pheophorbide b (1 µg/mL) or zanamivir (1 µM) at room temperature for 45 min. The mixture was subsequently treated with 0.3 mM 2′-(4-Methylumbelliferyl)-α-d-*N*-acetylneuraminic acid (MUNANA; Sigma-Aldrich), a substrate of NA, and incubated at 37 °C for 1 h. After the reaction was terminated with stop solution (11 mL of absolute ethanol, 2.225 mL of 0.824M NaOH), fluorescence was measured at excitation and emission wavelengths of 355 nm and 460 nm, respectively, using an EnSpire fluorescence analyzer (Perkin Elmer, Waltham, MA, USA).

### 4.11. Hemagglutination Inhibition Assay

IAV was serially diluted using 1X PBS. After incubation with chicken red blood cells (cRBCs, 0.5%) for 1 h at 4 °C, aggregation was confirmed. The lowest virus titer causing aggregation was 1 HA unit (HAU). For the experiment, 4 HAU was used. IAV was treated with DMSO, ethyl pheophorbide a (1 µg/mL), or ethyl pheophorbide b (1 µg/mL) with shaking at room temperature for 45 min. Following the addition of cRBCs (0.5%), aggregation was confirmed after 1 h at 4 °C.

### 4.12. Hemolysis Inhibition Assay

IAV (10^8^ PFU/mL) was treated with DMSO, ethyl pheophorbide a (1 µg/mL) or ethyl pheophorbide b (1 µg/mL) at room temperature for 1 h. Mixtures were incubated with cRBCs (2%) at 37 °C for 30 min, followed by sodium acetate–acetic acid (0.5 M, pH 5.2) at 37 °C for 30 min. After centrifugation at 800× *g* for 5 min, absorbance of the supernatant was measured at 560 nm.

## Figures and Tables

**Figure 1 molecules-28-00041-f001:**
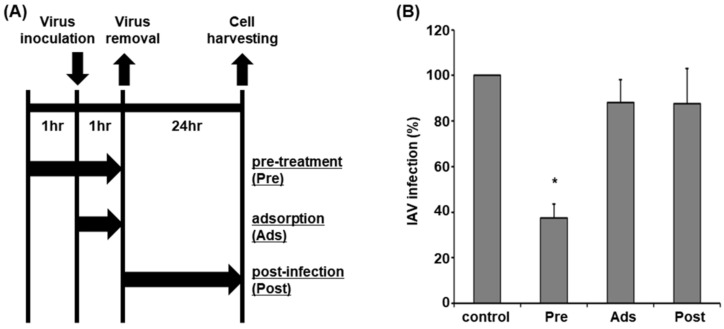
Virucidal activity of APE against IAV. The 70% ethanol extract of *A. pseudoglehnii* was used to treat IAV at a concentration of 10 µg/mL using three protocols. (**A**) Schematic representation of the time-of-drug-addition assay. (**B**) After treatments, viral RNA transcript levels were determined via qRT-PCR using primers specific for IAV M1 and GAPDH. Experiments were performed in triplicate and data expressed as GAPDH-normalized values relative to control. The viral RNA transcript level in DMSO-treated cells was set as 100%. * *p* < 0.05 (Student’s *t*-test). Pre, pre-treatment; Ads, Adsorption; Post, post-treatment.

**Figure 2 molecules-28-00041-f002:**
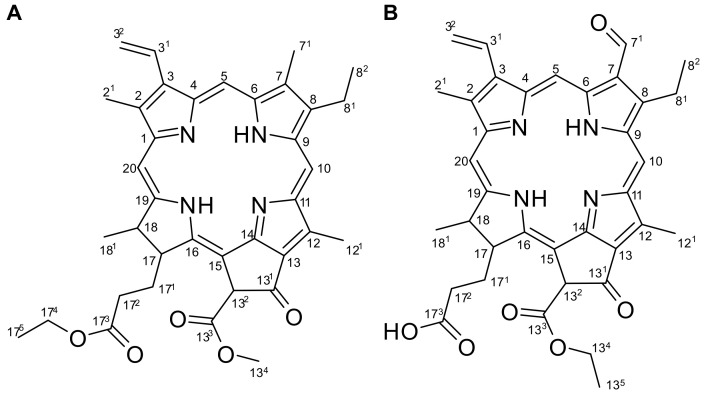
Ethyl pheophorbides a (**A**) and b (**B**) obtained from APE.

**Figure 3 molecules-28-00041-f003:**
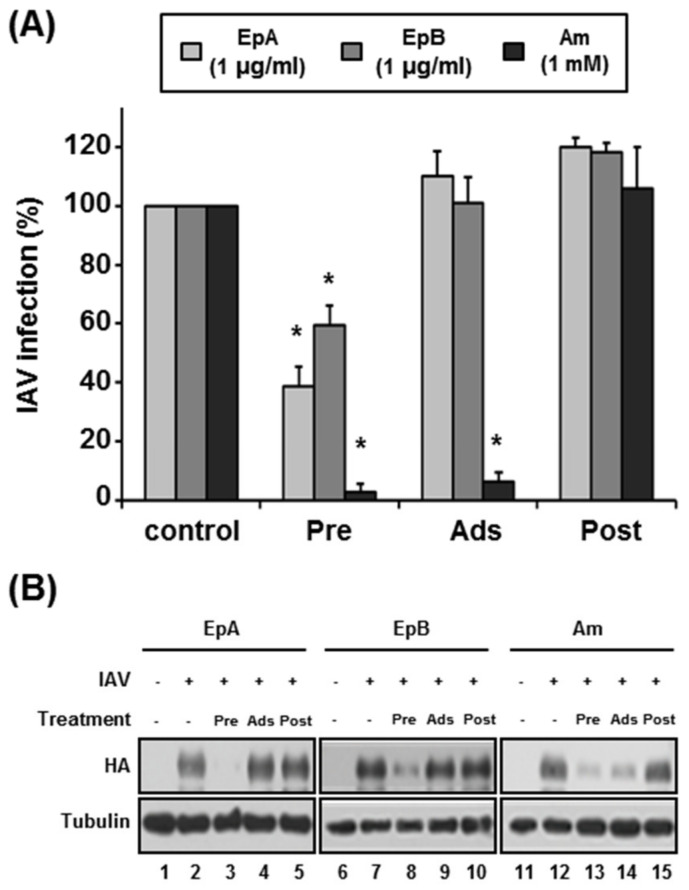
Virucidal activities of ethyl pheophorbides a and b. Following treatment of IAV with ethyl pheophorbide a (EpA) or b (EpB) at a concentration of 1 µg/mL, experiments were performed as described for Figure 1A. (**A**) Viral RNA transcript levels were measured via qRT-PCR using primers specific for IAV M1 and GAPDH. Amantadine (Am), a M2 ion channel inhibitor (1 mM), was used as the control. Experiments were performed in triplicate and data expressed as GAPDH-normalized values relative to control. The viral RNA transcript level in DMSO-treated cells was set as 100%. * *p* < 0.05 (Student’s *t*-test). (**B**) Equivalent amounts of whole cell extracts were subjected to western blot analysis with antibodies specific for IAV HA and tubulin. Pre, pre-treatment; Ads, Adsorption; Post, post-treatment.

**Figure 4 molecules-28-00041-f004:**
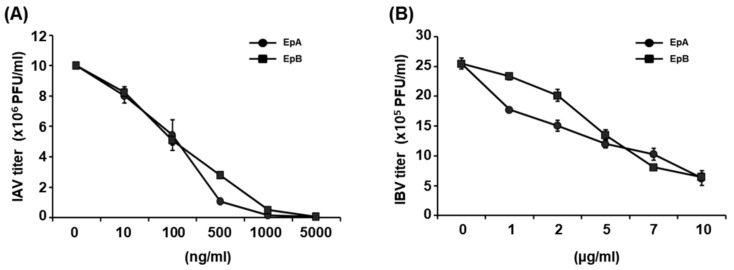
Inhibition of influenza virus infection by ethyl pheophorbides a and b. Serially diluted (**A**) IAV or (**B**) IBV were pre-treated with different concentrations of ethyl pheophorbide a (EpA) or b (EpB). After incubation at room temperature for 1 h, MDCK cells were infected with the mixture containing IAV and IBV at MOI of 0.01 and 0.001, respectively. At 1 h after infection, the mixture was removed and viral titers (PFU/mL) were determined 48 h after infection using a plaque assay. All experiments were performed in triplicate.

**Figure 5 molecules-28-00041-f005:**
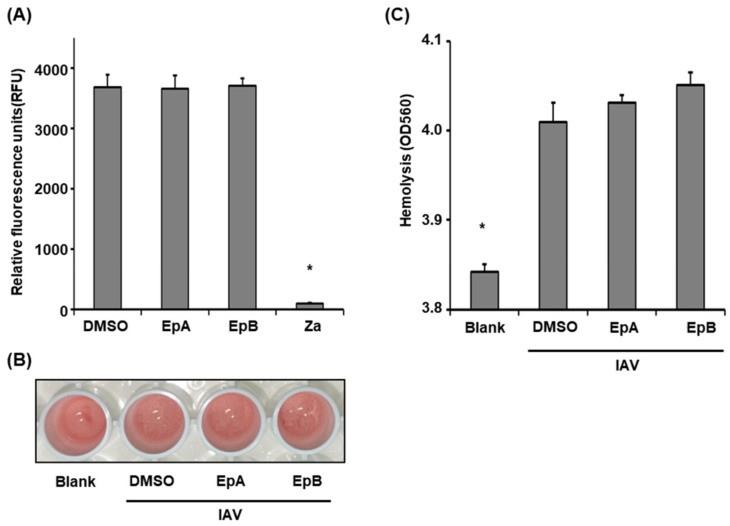
Effects of ethyl pheophorbides a and b on viral major surface glycoproteins. (**A**) IAV was treated with ethyl pheophorbide a (EpA, 1 µg/mL) or b (EpB, 1 µg/mL) for the NA inhibition assay. Zanamivir (Za, 1 µM), an NA inhibitor, was used as the control. (**B**,**C**) IAV was treated with ethyl pheophorbide a (1 µg/mL) or b (1 µg/mL) and HA inhibition was examined using (**B**) hemolysis inhibition and (**C**) hemagglutination inhibition assays. Experiments were performed in triplicate. * *p* < 0.05 (Student’s *t*-test).

**Figure 6 molecules-28-00041-f006:**
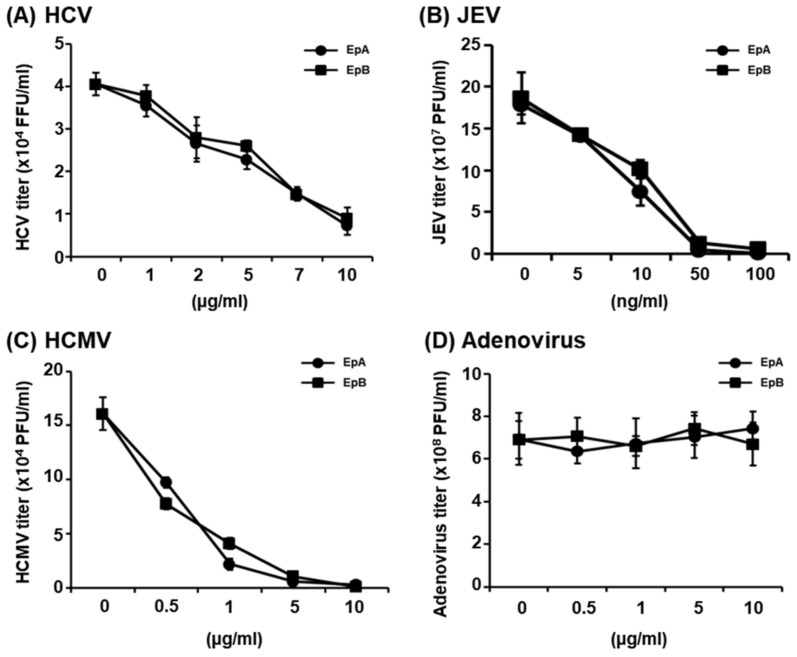
Virucidal activities of ethyl pheophorbides a and b against enveloped viruses. (**A**) HCV, (**B**) JEV, (**C**) HCMV and (**D**) adenovirus were pre-treated with different concentrations of ethyl pheophorbide a (EpA) or b (EpB) and incubated at room temperature for 1 h. Huh 7.5, HFF, BHK21 and 293A cells were infected with mixtures containing serially diluted (**A**) HCV, (**B**) JEV, (**C**) HCMV and (**D**) adenovirus, respectively. Viral titer was determined via immunofluorescence assay (for HCV) or plaque assays (for JEV, HCMV and adenovirus). Experiments were performed in triplicate.

**Table 1 molecules-28-00041-t001:** ^1^H and ^13^C NMR spectroscopic data on compounds **1** and **2** (*δ* in ppm, CDCl_3_, 500 and 125 MHz).

Position	1		2	
	*δ*_H_ Multi (*J* in Hz)	*δ* _C_	*δ*_C_ Multi (*J* in Hz)	*δ* _C_
1	–	142.1	–	143.6
2	–	131.9	–	132.2
2^1^	3.38 s	12.1	3.39 s	12.1
3	–	136.3	–	137.8
3^1^	7.96 dd (18.0, 11.5)	129.2	8.02 dd (17.5, 11.5)	128.7
3^2^	6.16 dd (11.5, 1.5)/6.27 dd (18.0, 1.5)	122.9	6.24 dd (11.5, 1.0)/6.38 dd (18.0, 1.0)	123.6
4	–	136.6	–	137.2
5	9.36 s	97.5	10.31 s	101.6
6	–	155.5	–	151.3
7	–	136.3	–	132.9
7^1^	3.20 s	11.2	11.17 s	187.8
8	–	145.2	–	159.5
8^1^	3.66 d (8.0)	19.5	3.90 q (7.5)	19.4
8^2^	1.65 t (8.0)	17.4	1.84 t (7.5)	19.1
9	–	150.7	–	147.2
10	9.50 s	104.5	9.68 s	107.0
11	–	138.0	–	138.0
12	–	129.1	–	128.7
12^1^	3.67 s	12.2	3.70 s	12.3
13	–	129.1	–	132.5
13^1^	–	189.7	–	189.5
13^2^	6.26 s	64.7	6.23 s	64.6
13^3^	–	173.0	–	169.3
13^4^	3.86 s	52.9	4.46 q (7.0)	66.9
13^5^	–	–	1.12 t (6.5)	14.1
14	–	149.7	–	150.7
15	–	105.3	–	104.9
16	–	161.4	–	164.0
17	4.20 ddd (9.0, 3.5, 2.0)	51.1	4.20 m	51.3
17^1^	1.24 br s/2.7 ddd (15.5, 9.5, 6.5)	29.8	2.23 m/2.50 m	29.6
17^2^	2.18 ddd (15.0, 10.0, 5.0)/2.47 ddd (16.0, 9.5, 6.5)	31.2	2.33 m/2.64 m	31.2
17^3^	–	172.3	–	174.0
17^4^	4.00 m	60.5	–	–
17^5^	1.09 t (7.0)	14.1	–	–
18	4.45 qd (7.5, 2.0)	50.1	4.46 qd (7.5, 2.0)	50.1
18^1^	1.80 d (7.5)	23.1	1.84 d (7.5)	23.1
19	–	169.6	–	172.8
20	8.56 s	93.2	8.55 s	93.4

## Data Availability

The data presented in this study are available in Appendix A.

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
