# Peer review of "Antiviral Activities of Ethyl Pheophorbides a and b Isolated from Aster pseudoglehnii against Influenza Viruses"

_molecules, 2022, doi:10.3390/molecules28010041_

Round 1
Reviewer 1 Report
Overall it is a well-written manuscript. The authors examined the antiviral activity of ethyl pheophorbides a and b. To be published in this journal, several questions need to be addressed.
Major points:
1. Compounds A and B only had function during the pre-treatment, indicating their blocking activity at the very beginning of the viral entry. However, in Figure.6 there were no positive hemagglutinin inhibition results. The authors should give an appropriate explanation to accommodate the previous inhibition results. By increasing the concentration of compounds A and B when doing HA or NA inhibition test, the authors may observe the difference.
2. Should further improve the discussion part. The authors can strengthen the discussion by: a. What the possible anti-influenza mechanism in this study is? b. Why there is an anti-viral activity difference between a and b from the structure analysis? c. More discussion about the antiviral activity against enveloped viruses?
Minor points:
1. There is only one figure in Figure.5. The authors can incorporate it into other figures or just put it in the supplementary materials.
2. Although most people know what are HCV, JEV, et al, the authors should give the full name at least when it first appeared in the manuscript.
Author Response
1. Compounds A and B only had function during the pre-treatment, indicating their blocking activity at the very beginning of the viral entry. However, in Figure.6 there were no positive hemagglutinin inhibition results. The authors should give an appropriate explanation to accommodate the previous inhibition results. By increasing the concentration of compounds, A and B when doing HA or NA inhibition test, the authors may observe the difference.
- As described in Figure 1A, viruses were directly treated with compounds before infection for the pre-treatment course. As shown in Figure 3, ethyl pheophorbides a and b were only effective when viruses were directly treated with them before infection. Drugs can directly interfere with viral infectivity by inactivating spike proteins and/or attenuating virion integrity. HA inhibition assays were performed appropriately because IAV induced hemolysis and hemagglutination (Figure 6B and C, compare blank with IAV). Currently, there are no effective FDA-approved HA inhibitors. We used concentrations up to 10 ug/mL, and there were no significant inhibitory effects against HA and NA.
2. Should further improve the discussion part. The authors can strengthen the discussion by: a. What the possible anti-influenza mechanism in this study is? b. Why there is an anti-viral activity difference between a and b from the structure analysis? c. More discussion about the antiviral activity against enveloped viruses?
- The antiviral mechanisms of ethyl pheophorbides a and b against enveloped viruses are discussed in lines 259-278. Similar to chlorophyll derivatives, ethyl pheophorbides a and b affect the viral membrane as photosensitizers.
- As described in lines 113-147 and figure 2, pheophorbides a and b are structurally different at positions 7, 13, and 17. In pheophorbide a, there is a methyl group (-CH3) at positions 7 and 13 (positions 71 and 134) and an ethyl group (-CH2CH3) at position 17 (positions 174 and 175). On the other hand, in pheophorbide b, there is a carbonyl group (-C=O) instead of a methyl group at position 7, and an ethyl group instead of a methyl group at position 13. Also, there is a hydroxyl group (-OH) instead of an ethyl group at position 17. However, structural differences do not play significant roles in antiviral activities of ethyl pheophorbides a and b because the IC50 values are not significantly different. Determination of the critical moiety is the focus of future study.
- Antiviral activities of ethyl pheophorbides a and b are discussed in lines 267-278.
3. There is only one figure in Figure.5. The authors can incorporate it into other figures or just put it in the supplementary materials.
- As suggested by the reviewer, Figure 5 has been moved into the supplementary materials.
4. Although most people know what are HCV, JEV, et al, the authors should give the full name at least when it first appeared in the manuscript.
- The manuscript has been revised as suggested by the reviewer (lines 232-234).
Reviewer 2 Report
The article is well-designed. Introduction part is also well-composed and meets all requirements. Anyway, in the "Results" part there are some points to be changed and improved, namely:
1. Line 82-92 The description of methodological approach should be transferred to "Materials and Methods" section. In the "Results" please leave only experimental data and their description. Authors can provide also a short description of the groups whithout detalizing the experimental approach.
2. Line 76-109. The description of methodological approach in the Fidure 1 title should be removed to "Materials and Methods" section.
3. Please describe why you are using 10 mkg/ml concentration of APE?
Author Response
1. Line 82-92 The description of methodological approach should be transferred to "Materials and Methods" section. In the "Results" please leave only experimental data and their description. Authors can also provide a short description of the groups whithout detalizing the experimental approach.
- The manuscript has been revised as suggested by the reviewer (lines 83-95, 373-383).
2. Line 76-109. The description of methodological approach in the Figure 1 title should be removed to "Materials and Methods" section.
- The manuscript has been revised as suggested by the reviewer (lines 100-108, 373-383).
3. Please describe why you are using 10 ug/ml concentration of APE?
- For initial screening, two different concentrations (10 and 100 ug/mL) of APE were used. APE exhibited no cytotoxic effects against MDCK cells up to 100 ug/mL. The data with 10 ug/mL is a representative sample of the study.
Reviewer 3 Report
In this work, the ethanol extracts from A. pseudoglehnii against influenza virus was evaluated. The authors have demonstrated that two chlorophyll derivatives, ethyl pheophorbides a and b, isolated as active components of APE, exerted virucidal effects with no evident cytotoxicity. This manuscript was important. However, publication of this manuscript in its present form is not recommended. To be considered further for publication, this work will need further clarifications in support of the claims made in the paper. Therefore, my opinion was revise and resubmit.
Some specific points of concern were noted below:
1. The main problem is that the authors mention that the compound is virucidal. However, the experiments in the paper do not prove this.
2. The use of natural products to against IAV was not described in introduction. Authors should improve it because that was necessary.
3. Whether APE is toxic to cells? What is the safety concentration of APE? How to determine the concentration of 10 µg/mL? Author should describe it in the manuscript, rather than ignore it.
4. Line 82-94: This section mainly reflects the anti-influenza virus activity of APE, and the plethora of methods should be carried out at “4. Materials and Methods”.
5. Add “Immunofluorescence Assay” in the “2. Results”.
6. How to determine the concentration of EPA and EPB (1µg/L)?
7. Line 173: What is the basis of “IAV and IBV at MOI of 0.01 and 0.001”?
8. Line 252-285: The discussion was too simple. There was no current application of natural products in anti-influenza virus, and the similarities and differences between EPA, EPB and other natural antiviral products. No reasonable explanation of each parallel experiment and conclusion was lacking. There were no enough references to support.
9. Line 317: “Upon ≥ 50% cell death”, how to judge?
10. Most references were published too long.
Author Response
1. The main problem is that the authors mention that the compound is virucidal. However, the experiments in the paper do not prove this.
- As shown in Figure 1 and 3, APE as well as ethyl pheophorbides a and b was only effective when viruses were directly treated with them before infection. Direct treatment of eneveloped viruses including influenza viruses with APE and ethyl pheophorbides a and b significantly inhibited viral infectivity. On the other hand, they had no effects against non-enveloped viruses. APE and ethyl pheophorbides a and b affect the viral membrane and inactivate enveloped viruses. Hence, we used the term, virucidal, as these compounds inactivate viruses.
2. The use of natural products to against IAV was not described in introduction. Authors should improve it because that was necessary.
- As suggested by the reviewer, natural products against influenza viruses were described in introduction (lines 70-73).
3. Whether APE is toxic to cells? What is the safety concentration of APE? How to determine the concentration of 10 µg/mL? Author should describe it in the manuscript, rather than ignore it.
- For initial screening, two different concentrations (10 and 100 ug/mL) of APE were used. APE exhibited no cytotoxic effects against MDCK cells up to 100 ug/mL. The data with 10 ug/mL is a representative sample of the study. The manuscript has been revised to include the information (lines 94-95).
4. Line 82-94: This section mainly reflects the anti-influenza virus activity of APE, and the plethora of methods should be carried out at “4. Materials and Methods”.
- The manuscript has been revised as suggested by the reviewer (lines 83-95, 373-383).
5. Add “Immunofluorescence Assay” in the “2. Results”.
- Immunofluorescence assay was used to determine HCV titer in Figure 7.
6. How to determine the concentration of EPA and EPB (1µg/L)?
- As in Figure 4, we used different concentrations of EpA and EpB to determine IC50 values of these compounds. At 1ug/mL, both compounds significantly inhibited IAV infection. Thus, we used 1ug/mL of compounds to further studies.
7. Line 173: What is the basis of “IAV and IBV at MOI of 0.01 and 0.001”?
For influenza virus infection, MOIs of 0.1-0.0001 are used. To yield similar infection rates, appropriate MOIs must be primarily determined in different experimental settings. Cells were infected with various MOIs of IAV and IBV, and MOIs exhibiting similar infection rates were determined by observing cytopathic effects (CPE). In our experimental settings, we found that MOIs of 0.01 and 0.001 for IAV and IBV, respectively, generate similar infection rates.
8. Line 252-285: The discussion was too simple. There was no current application of natural products in anti-influenza virus, and the similarities and differences between EPA, EPB and other natural antiviral products. No reasonable explanation of each parallel experiment and conclusion was lacking. There were no enough references to support.
- As suggested by the reviewer, natural products against influenza viruses were described in introduction (lines 70-73). The comparison of ethyl pheophorbides with other natural products exhibiting virucidal activities is discussed in lines 267-277.
9. Line 317: “Upon ≥ 50% cell death”, how to judge?
Adenovirus generation causes 293A cell death which is determined by observing CPE (cell detachment and rounding). The manuscript has been revised to clarify the information (line 319).
10. Most references were published too long.
-As suggested by the reviewer, previous references #2, 3, 4, and 13 were replaced.
Round 2
Reviewer 2 Report
All changes made by author are appropriate.
Author Response
All changes made by author are appropriate.
- We appreciate the reviewer's comments.
Reviewer 3 Report
Our main concern of this article is that this one test does not prove that the compound is a virucidal agent. Maybe the title of this article could be changed.
Author Response
Our main concern of this article is that this one test does not prove that the compound is a virucidal agent. Maybe the title of this article could be changed.
- As suggested by the reviewer, the title of the article has been changed (virucidal to antiviral).